

# Observations of ozone depletion events in a Finnish boreal forest

Xuemeng Chen[1], Lauriane L. J. Quéléver[1], Pak L. Fung[1], Jutta Kesti[2], Matti P. Rissanen[1], Jaana Bäck[1],
Petri Keronen[1], Heikki Junninen[1], Tuukka Petäjä[1], Veli-Matti Kerminen[1] and Markku Kulmala[1]

[1] Division of Atmospheric Sciences, Department of Physics, FI-00014 University of Helsinki, Finland
[2] Atmospheric Composition Research, Finnish Meteorological Institute, P.O. Box 503, FI-00101 Helsinki, Finland

*Correspondence to:* Xuemeng Chen (xuemeng.chen@helsinki.fi)

**Abstracts.** We investigated the concentrations and vertical profiles of ozone over a 20-year period (1996-2016) at the
SMEAR II station in Southern Finland. Our results showed that the typical daily median ozone concentrations were in the
range of 20 – 50 ppb with clear diurnal and annual patterns. In general, the profile of ozone concentrations illustrated an
increase as a function of heights. The main aim of our study was to address the frequency and strength of ozone depletion
events at this boreal forest site. We observed more than a thousand of 10-min periods at 4.2 m, with ozone concentrations
below 10 ppb, and a few tens of cases with ozone concentrations below 2 ppb. Among these observations, a number of
ozone depletion events that lasted for more than 3 hours were identified, and they occurred mainly in autumn and winter
months. The low ozone concentrations were likely related to the formation of a low mixing layer under the conditions of low
temperatures, low wind speeds, high relative humidities and limited intensity of solar radiation.

## 1. Introduction

Ozone ($O_3$) is one of the most important compounds in atmospheric chemistry (e.g. Monks, 2005; Monks et al. 2015). It is a
strong greenhouse gas and a key contributor to the oxidation capacity of the atmosphere (Toumi et al., 1994; Simpson et al.
2007). Ozone is also the principal source of the essential atmospheric oxidants, being the major photochemical precursor of
the OH radical (Atkinson and Arey, 2003) and a source of the nocturnal $NO_3$ radical through the $NO_2 + O_3 \rightarrow NO_3$ reaction
(Brown and Stutz 2012). Very recently, the importance of ozone in initiating the autoxidation of volatile organic compounds
(VOCs) in the gas-phase was recognised, producing rapidly a range of highly-oxidised multifunctional compounds (HOMs,
(Ehn et al., 2014; Rissanen et al., 2014)), which can participate in the formation of secondary organic aerosols and/or
contribute significantly on their early growth (Kulmala et al., 1998; Ehn et al., 2014; Trostl et al., 2016). While stratospheric
ozone acts to shield the Earth from detrimental ultraviolet radiation, the tropospheric ozone is harmful for human health (e.g.
Shang et al., 2013; Turner et al., 2016). It causes negative consequences throughout the biosphere (Gregg et al., 2003;
Ashmore, 2005) and reduces crop yields (Emberson et al., 2001; Feng and Kobayashi, 2009), when it is either deposited to
the surface of or taken up and metabolised by plants.

The concentration of tropospheric ozone is governed by its sources and sinks. In the troposphere, $O_3$ is produced via
complex photochemical cycles involving VOCs and nitrogen oxides ($NO_x$ consists of NO and $NO_2$) (Sillman et al., 1990;
Sillman, 1999). Transport from the stratosphere brings additional ozone to its tropospheric budget. The loss processes of
tropospheric ozone include dry and wet deposition, photolysis and reactions with other trace gases. These contributions form
an equilibrium state that maintains tropospheric ozone-mixing ratios at concentration levels of about tens of parts per billion
(ppb) (Vingarzan, 2004; Monks et al., 2015). Although tropospheric ozone is relatively well mixed in a broad sense, its
concentration can vary notably with both time and space depending on the prevailing meteorological conditions. Regional
and local surface $O_3$ concentrations are affected, for example, by horizontal and vertical advections of pollutants,



entrainment from the free troposphere and other boundary layer dynamic processes (Simpson et al., 2007; Cao et al., 2016; Tang et al., 2017; Wang et al., 2017; Zhou et al., 2017)

The most well-known ozone depletion phenomena are the stratospheric "ozone-hole" episodes taking place above the polar
regions caused by the catalytic halogen chemistry (Molina and Rowland, 1974; Solomon, 1999), as well as the frequently-reported $O_3$ depletion events in a polar planetary boundary layer (Simpson et al., 2007). In addition, significant decreases in local surface $O_3$ concentrations have been observed in polluted urban environments (Gregg et al., 2003). While in the polar areas halogenated compounds are responsible for most of the surface ozone destruction, in urban locations low local ozone concentrations are typically related to complex atmospheric chemistry involving anthropogenic pollutants,
mainly nitrogen oxides (Gregg et al., 2003; US Nat. Res. 1992). Although anormal reductions in surface ozone concentrations have been observed at different geographical locations, there lacks a general characterisation of the vertical features of such $O_3$ depletion phenomena. Thus, it is imperative to study the vertical profile of $O_3$ to provide insights into observations of surface ozone depletion.

Based on previously reported depletion events in surface ozone, we investigated in detail ozone concentrations and profiles using long-term (>20 years) data collected between 1 January 1996 and 30 June 2016 at the Hyytiälä SMEAR II station, a boreal forest site in Finland. Our main goals in this study were 1) to characterise the typical seasonal and diurnal behaviour of vertical ozone concentration profiles within and above the boreal forest, 2) to evaluate statistically the low ozone concentration episodes, 3) to assess the atmospheric conditions in association with these low concentration episodes, and 4)
to identify key atmospheric features during the ozone depletion events.

## 2.   Experimental setup and data analysis

The SMEAR II (Station for Measuring forest Ecosystem-Atmosphere Relation) infrastructure, located in a boreal forest at Hyytiälä, Southern Finland (61°51'N, 24°17'E, 181 meters above sea level), offers a unique set of information based on
continuous measurements of atmospheric composition and meteorological variables (Hari and Kulmala, 2005). These measurements are located at several heights on a mast, ranging from the below-canopy level (4.2 m, 8.4 m and 16.8 m) to the above-canopy levels (33.6 m, 50.4 m, 67.2 m, 101 m and 125 m), capturing the vertical profile of the lower atmosphere at this site.

The forest stand is dominated by Scots pines (*Pinus sylvestris* L.) with an average height of 16-18 m (living canopy height ca. 10-12 m) and age of 55 years. Other, less abundant species include Norway spruce (*Picea abies* L. H. Karst.), silver birch (*Betula pendula* Ehrh.) and trembling aspen (*Populus tremula*) which together form ca. 25% of the biomass (Ilvesniemi et al., 2009). The ground vegetation consists of dwarf shrubs (e.g. *Vaccinium* sp., *Calluna vulgaris*) and mosses (*Pleurozium schreberi*, *Dicranum* sp.). For most of the vertical measurement levels, the data were available since January 1996. But the
highest measurement levels (101 m and 125 m) were introduced in February 2013. These data altogether form a continuous dataset of atmospheric observations over a 20-year period.

### 2.1 Trace gas measurements

All the trace gas concentrations were recorded in every minute at all the aforementioned measurement levels. $O_3$ concentrations were determined by a photometric TEI 49C ozone monitor (Thermo Fisher Scientific, Waltham, MA, USA). The instrument had a detection limit of 1 ppb and a relative accuracy of ± 3 %.

Carbon monoxide (CO) and carbon dioxide ($CO_2$) concentrations were quantified by their infrared (IR) absorption
characteristics. CO was measured with a Horiba APMA 360 monitor (Horiba, Kyoto, Japan) with a detection limit of 20 ppb


and relative accuracy of ±3 %. $CO_2$ was measured with an URAS 4 $CO_2$ monitor (Mannesmann Hartmann & Braun, Frankfurt am Main, Germany) that had a relative accuracy of ±1 % The $CO_2$ signal was corrected for $H_2O$ vapour interference.

Nitrogen oxide ($NO_x$ consists of NO and $NO_2$) concentrations were measured with a TEI 42CTL photometric analyser (Thermo Fisher Scientific, Waltham, MA, USA). The measurement had a 0.1 ppb limit of detection during 2006 measurements and this was increased to 0.15 ppb for the measurements conducted in 2009 due to modification of the instrument. The relative accuracy in all of the NOx measurements was ±10%.

**2.2 Ancillary measurements**

The temperature was monitored from all the heights, up to 67.2 m, with custom-made 4-wired PT-100 sensors. The measurement spots were shielded in order to avoid interferences due to solar radiation and ventilated by fans to ensure a homogeneity in the measured air parcel. The accuracy of the temperature measurement was estimated to ± 0.2 °C via a
comparison with a mercury-based thermometer.

From April 1998 to mid-June 2012, the Relative Humidity (RH) values were obtained via measurement of dew point temperature and air temperature at 16.4 m on the sampling tower using a chilled mirror hygrometer (DewTrak Model 200M Meteorological Humidity System, EdgeTech, Marlborough, MA, USA.). The measurement of the dew point and ambient
temperatures were retrieved with an accuracy of ± 0.3 °C given a RH accuracy of ± 3 %. Since 16 June 2012, the relative humidity is directly measured with an RH sensor (Rotronic MP102H with Hygroclip HC2-S3, Rotronic AG, Bassersdorf, Switzerland), providing accuracy of ± 0.8 %.

The total global radiation was calculated as the sum of direct and diffuse solar radiations. The measurement was placed on a
separate tower next to the mast at the height of 18 m. The radiation was measured using pyranometers: a Reemann TP 3 pyranometer, (Astrodata, Tõravere, Tartumaa, Estonia) from January 1996 to mid-June 2008, and a Middleton Solar SK08 First Class Pyranometer (Middleton Solar, Yarraville, Australia) since 15 June 2008. These devices have a detection limit of 5 W/m². The irradiance of solar radiation on a hemispherical surface (sun disk) was monitored in a wavelength range of 0.30 - 4.8 $\mu$m, with an accuracy of ± 5 %.

Wind speeds at different levels of the mast were measured either with cup anemometers or ultrasonic anemometers, or both, with an accuracy of ± 0.1 m/s. Cup anemometers (A101M/L, Vector Instruments, Rhyl, Clwyd, UK) were used for monitoring wind properties at 4.2 m, 8.4 m, 16.8 m, 33.6 m 50.4 m and 74 m from January 1996 to September 4th, 2003. Thereafter, the cup anemometers were replaced at all the measurement levels by ultrasonic anemometers (Ultrasonic
anemometer 2D, Adolf Thies GmbH, Göttingen, Germany). Since 7 July 2011, the wind measurements at 74 m were moved down to 67.4 m. Two 3D ultrasonic anemometers (Metek uSonic-3, Elmshorn, Germany) were installed at 67.4 m and 125 m in November 2015.

Friction velocity data were used to characterise the turbulent conditions. The friction velocity ($u_*$) was determined as 30-min
averages of square root of the covariance between horizontal and vertical winds (Monteith and Unsworth, 2013),

$$u_* = \sqrt{\overline{u'w'}}. \tag{1}$$

$u'$ and $w'$ are the fluctuations in the horizontal and vertical winds, respectively. Horizontal and vertical winds were measured at 23.3 m with a fast-response acoustic anemometer (Solent HS1199) on the radiation tower.

For assessing the stability of the atmosphere, the potential temperatures were calculated using the following expression (e.g. Monteith and Unsworth, 2013 )

$$\theta = T \left(\frac{P_0}{P}\right)^{R/c_P}, \tag{2}$$

where $T$ is the ambient temperature (in Kelvins), $P_0$ is 1000 mbar, $P$ is the ambient pressure (in mbar), $R$ is the gas constant,



and $c_p$ is the isobaric heat capacity. For air, the ratio gas constant to isobaric heat capacity ($R/c_p$) typically has a value of 0.286.

### 2.3 Data processing and handling

With the exception of wind data, both gas and meteorological data were arithmetically averaged into a 10-min time resolution. The wind data were treated as vectors, where the arithmetic averaging was applied to the two horizontal wind components of the decomposed vectors and the 10-min wind data were restored from these averaged components.

The duration of consecutive low ozone concentrations was determined based on the 10-min data for the concentrations below threshold values of 10, 5, 2 and 1 ppb. Occasionally, the data were available only for every second 10-min interval, mainly between Jan. 1996 and Feb. 1997 for all the heights and at heights of 101 m and 125 m from Feb. 2013 onwards. A gap-filling procedure was applied to such data in order to estimate the duration of a low ozone episode: if ozone concentrations were found below a threshold in every second 10-min interval with the value in between being missing, then the missing value in between two low concentration records was treated as a low concentration point. Hereafter, the low ozone episodes are referred as ozone concentration drops, and when the ozone concentration remained below a given threshold (10, 5, 2, or 1 ppb) for more than 30 min, they are considered as ozone depletion events.

### 3. Results and Discussions

#### 3.1. Diurnal and seasonal variations in ozone profiles

A clear seasonality was seen in ozone concentrations at different heights (Figure 1). The ozone concentrations were the highest in spring (March-April) at all heights and then declined towards the lowest values seen in early winter (November). This observation is similar to the reported feature in surface ozone concentrations at several northern mid-latitude sites (Oltmans et al., 2006; Parrish et al., 2013). Atlas et al. (2003) and Vingarzan (2004) pointed out that the spring peak is probably caused by spring recovery of photochemical production (Dibb et al., 2003) and by ozone accumulation through winter (Liu et al., 1987) at the Northern mid to high latitudes, but hardly by the transportation from stratosphere, even though the intrusion of ozone from stratosphere has been reported to be a major source for springtime ozone appearing in the middle troposphere at high northern latitudes (Dibb et al., 2003). Notably, starting from early summer and continuing until late autumn, ozone concentrations tended to increase with an increasing measurement height, being clearly the lowest close to the surface (Figure 1). This feature is probably linked to some combination of the biotic ozone uptake by vegetation, ozone deposition on surfaces inside the forest and ozone loss through chemical reactions with forest emissions (Rannik et al., 2012; Monks et al., 2015; Clifton et al., 2017).

The diurnal behaviour of the vertical ozone concentration profile was similar in spring (March-May) and summer (June-August) (Figure 2): a high ozone concentration was seen throughout the vertical extend of the measurements between about 10:00 and 20:00, likely related to the photochemical production of ozone. The highest ozone concentrations were generally seen above the canopy level (at about 50-60 m) during the daytime, with a clear decline with the decreasing height. This observation can be attributed to the loss of ozone by the forest (Rannik et al., 2012; Monks et al., 2015). During the evening in spring and summer, low ozone concentrations were first (at around 20:00) seen close to the ground and then later on gradually at higher up, so that the diurnal minimum ozone concentration was reached between about 05:00 (4.2 m) and 08:00 (125 m). This phenomenon results from the ozone deposition at night, when the photochemical production of



ozone is prohibited. Ozone could also be consumed through oxidation reactions with biogenic organic compounds that are accumulated in the nocturnal boundary layer (Hakola et al., 2012). After sunrise, turbulent mixing in the boundary layer is initiated, leading to an increase in the mixed layer height and entrainment of ozone into this layer from above, in addition to which ozone starts to form via photochemical reactions. As a consequence, the ozone concentrations at all levels recovered
rapidly, with a tendency of the concentration recovering at higher levels prior to that at lower levels.

The autumn ozone concentration pattern followed that seen in spring and summer, but with a slower ozone concentration recovery in the morning (Figure 2). Also, the onset of the concentration recovery occurred slightly later in autumn (at 9:00) than in spring and summer (between 6:00-8:00). These features are very likely related to the shorter daytime length in
autumn compared with spring and summer (Figure 3). The later sunrise time in autumn facilitated a prolonged deposition-dominant ozone budget and a delayed replenishment. The reduced solar radiation intensity in autumn was unable to sustain a rapid enrichment of ozone from the residual layer. Meanwhile, the photochemical production of ozone was also weakened. As a consequence, the autumn ozone profile showed a gradual concentration recovery from high to low measurement levels in the morning. Moreover, the occurrence of the ozone concentration declination in the afternoon in
autumn at around 17:00, well before that in spring and summer, can be possibly ascribed to the earlier sunset time and corresponding increase in the surface layer stability. Finally, the highest ozone concentrations tended to shift upwards in heights from spring to autumn. No clear diurnal pattern in ozone concentrations was seen in winter.

### 3.2.     Frequencies of low ozone concentration episodes

As described before, we defined four low-concentration thresholds (10, 5, 2 and 1 ppb) to characterise low ozone concentration episodes. When such an episode lasted for more than 30 min, it is considered as an ozone depletion event, otherwise as an ozone drop. To assist the characterisation of long-lasting depletion events longer than 3 hours, a gap-filling procedure was utilised as described in Section 2.3.

The observation frequency, i.e. the number of observations, of ozone concentration drops below 10 ppb exhibited similar
diurnal variations at the different measurement levels (Figure 4). The most pronounced diurnal variation was seen inside the canopy layer ($\leq$ 16.8 m), with low ozone concentrations frequently observed around at 4:00 - 8:00 (Figure 4a, b and c). Low ozone observations were rare during the noon. The diurnal behaviour of ozone concentration drops below 5 ppb resembled those of the below 10 ppb drops. However, excluding the 4.2-m measurement level, ozone concentrations below 5 ppb occurred also often in the afternoon at around 16:00 - 20:00. For the occurrence of ozone concentration drops below 2
ppb, in addition to the expected noon low, the diurnal distribution exhibited a second trough at around 2:00 in the early morning prior to the morning high, again except at 4.2 m. Interestingly at 4.2 m, the moment at which the morning peak observations occurred shifted in time from 4:00 to 6:00 for the sequence of ozone concentration drops below 10, 5 and 2 ppb. This feature was also perceptible at 8.4 m and 16.8 m (Figure 4b and c). No systematic diurnal variations were recognised for the observation frequency of ozone concentration drops below 1 ppb.

Ozone depletion events that lasted for more than 30 min occurred mainly in autumn and winter (Figure 5). March had the lowest number of ozone depletion events below 10 ppb, after which the number of these events tended to increase over the course of both months (towards December) and measurement heights (from low to high). Except in May and June, ozone depletion events below 10 ppb were observed to reach the measurement level of 67.2 m. Ozone depletion events below 5
ppb were mainly observed below the canopy level ($\leq$ 16.8 m) during June–September, while only during October –January and in March they reached 67.2 m. In February, April and May, no ozone depletion events below 5 ppb were observed at any of the measurement levels. The below-2-ppb ozone depletion events exhibited a similar feature to the below-5-ppb events, however with the depletion seen only at the lowest measurement level (i.e. 4.2 m) in January and summer months. No depletion events below 2 ppb were observed during February–May. The below-1-ppb depletion events concentrated in the
period of October to December.





We found a number of long-lasting (>3 hours) ozone depletion events (Table 1). During 1996 - 2016, there were 4 such events, in which the ozone depletion below 2 ppb was observed up to the height of 67.2 m. This number increased to 26 in the case of below-5-ppb depletion events and to 161 (or 148 without the gap-filling procedure) in the case of below-10-ppb events. Long-lasting ozone depletion events below 10 ppb occurred all year around, whereas no such events below 5 ppb was observed in February, April, May or June (Figure 6). Long-lasting ozone depletion events below 2 ppb were found in October, November and December, whereas those below-1-ppb events occurred only in October and November and they were generally associated with high humidity conditions.

The variations of median diurnal ozone concentrations for different months derived from all 10-min data ranged from 17 to 45 ppb, with diurnal cycles seen in almost all months, except in January and December (black lines in Figure 6). The median ozone concentrations obtained from the days with observations of long-lasting ozone depletion events were, in general, lower than those derived from all 10-min data, and displayed similar diurnal patterns to those derived from all 10-min data during May–October. No systematic diurnal behaviour was identified for any types of ozone depletion events between November and April, nor was it found for the below-2-ppb and the below-1-ppb depletion events in October. The difference in the median ozone concentrations determined between on all the days and on depletion event days was about 10-15 ppb between October and February. In July-September, this concentration difference was big in the early morning, which then diminished towards noon and enlarged again after 16:00, as a result of boundary layer dynamics.

### 3.3. Features in low ozone episodes

Ozone concentrations were, on average, lower at higher relative humidities (Figure 7a), and low ozone episodes were only observed under high relative humidity (> 70 %) conditions. According to Altimir et al. (2006), moisture in the air enhances the total ozone deposition, possibly due to the high solubility of ozone in water (Sotelo et al., 1989). Thus, moisture content in the air is likely to be one of the key factors controlling these low ozone episodes. Low ozone concentrations occurred mostly during the night-time, associated with little global radiation (Figure 7b) and low wind speeds (Figure 7d). Air temperatures have been reported to have a minor effect on the total ozone deposition (Rannik et al., 2012). Although we observed no direct relationship between the air temperatures and ozone concentration drops (Figure 7c), the temperature might indirectly affect ozone deposition by influencing e.g. the relative humidity.

When ozone concentration dropped lower than 1 or 2 ppb at 67.2 m, there was typically a very small ozone concentration difference between the 67.2-m and 4.2-m heights, i.e. no gradient in the vertical ozone concentration profile. These episodes were found in association with friction velocities between 0 and 1 m/s, which indicates relatively stable conditions with low vertical mixing. Small ozone gradients were mainly observed at low friction velocities (Figure 8a). This is consistent with the observation, depicted in Figure 7b, that most of the low ozone concentrations occurred under relatively dark conditions with low solar radiation. Ozone concentration below 5 or 10 ppb observed at 67.2 m occurred mostly when the potential temperature difference between the 67.2 m and 4.2 m was around 0 °C (Figure 8b), indicating neutral atmospheric conditions. Under such conditions, the atmosphere does not favour large-scale vertical mixing of ozone, yet small-scale turbulent eddies could be formed, aiding the deposition of ozone and leading to the occurrence of ozone concentration drops.

In addition, there existed a number of cases, where the 4.2-m ozone concentration was low while the concentration difference between 67.2 m and 4.2 m was large (Figure 9a), i.e. a gradient in the vertical ozone concentration profile. These episodes were also observed at very low friction velocities (Figure 8c), but they tended to exhibit larger concentration differences at larger differences in the potential temperature between 67.2 m and 4.2 m (Figure 8d). Such observations likely indicate that the gradients in the vertical ozone concentration profile occurred under calm and stably stratified conditions. The narrow range of the low friction velocities (0-0.5 m/s) observed during these cases may suggest a nearly invariant deposition sink for ozone. In such circumstances, the smaller the mixing height, the more effective loss of surface ozone can be expected, due to the lack of ozone replenishment from higher up, which possibly results in the occurrence of the large



number of ozone depletion events only at lower heights (over 200 below-10-ppb events at the three lowest heights, Table 1).

Taken the above-described features in the vertical ozone concentration profile as the basis, two major types of ozone depletion events were identified: 1) the ozone depletion occurred at low measurement levels only with a clear gradient in the vertical ozone profile, termed here as near-surface depletion, and 2) the ozone depletion extended to 67.2 m with a negligible ozone concentration difference between 67.2 m and 4.2 m, termed here as full-profile depletion (Figure 9 a, c and d). A possible third type lies in between the near-surface and full-profile depletion, in which decreases in ozone concentration could be seen at high measurement levels but with a remarkable difference in the ozone concentration between 67.2 m and 4.2 m and a clear gradient can be seen through the vertical concentration profile. This third type is referred as gradient-profile ozone depletion. However, the boundary between the gradient-profile and the near-surface depletion can often be vague in ambient data, i.e. a depletion event classified as a near-surface event may be re-categorised into the gradient-profile type if a longer period is considered when a decrease trend in high level ozone concentrations can be seen. During years 1996-2016, over a hundred full-profile below-10-ppb depletion events were observed, twenty-six below-5-ppb events, four below-2-ppb events, and even one below-1-ppb event (Table 2). Presumably, we define that a near-surface depletion event should fulfil two criteria: 1) the 1st quartile of the difference in the ozone concentration between 67.2 m and the height that the depletion reaches has a value larger than 2 ppb, and 2) the 3rd quartile of the ozone concentration measured at 67.2 m during the depletion event is larger than 10 ppb. Then seven below-10-ppb events that last longer than 3 hours with depletion reaching 33.6 m could be identified during our study period (Table 2).

The near-surface and the gradient-profile ozone depletion events could be observed throughout the year, except in spring, with more pronounced ozone gradients typically seen in later summer and autumn (Figure 9 b). However, the full-profile depletion events were primarily observed in winter months (Figure 9 b). The near-surface depletion events, though often observed at night and early morning hours under relatively dark conditions, were not exclusively restricted in this time window. Occasionally, they also occurred during daytime hours, probably when a stable condition prevails resulting in a low mixing height.

An analysis of atmospheric conditions was carried out for a near-surface depletion event encountered during the night of 14-15 September 2009 (Figure 9c), and for a full-profile depletion event that took place on the morning of 23 October 2006 (Figure 9d). The near-surface depletion event was initiated before the sunset on 14 September and lasted over the course of the night. However during the 2-3 hours prior to the sunrise, unlike the ozone concentrations at 4.2 m and 8.4 m showing a continuous decrease, the ozone concentration at 16.8 m (about the forest canopy height) tended to level out. This observation is likely ascribed to the formation of two decoupling layers above and below the forest canopy. Upon the sunrise, the ozone concentration at 4.2 m and 8.4 m started shortly to recover from their lowest levels, while an instantaneous concentration recovery was possibly seen at 16.8 m, being accompanied by a decrease in the 33.6-m ozone concentration. These phenomena indicate a rapid development of turbulent mixing vertically above the canopy level. But the break-up of the decoupling and the extension of the mixing height to 67.2 m came with some time lags. Interestingly, after the ozone concentration levels at different heights converged at around 9:00, there seems to be a transient obstruction in the further increase of ozone concentrations below 16.8 m. The ozone concentrations at 16.8 m and higher up nearly levelled out until 10:00 while the 8.4-m and 4.2-m concentrations showed a drop before a concentration increase was perceptible. Such a feature (in some cases, even a clear reduction in the ozone concentrations measured at all heights) can be seen in more than 30% of the below-10-ppb near-surface depletion events at 4.2 m. These phenomena might be attributed to the ozone loss via photolysis and other consumptions through chemical reactions. In contrast, the full-profile event on 23 October 2006 occurred before the sunrise and lasted through the morning hours, even until almost 14:00 at 4.2 m. The recovery of the ozone concentrations was first seen at 67.2 m before noon and then gradually at lower heights.

Both of near-surface and full-profile depletion events displayed distinct features in relation to the air temperature (Figure 10 a-l). While both depletion events were observed at low temperature conditions, a temperature gradient was only present during the near-surface depletion event, with higher temperatures found at the highest measurement levels. This observation





shows that this near-surface depletion event occurred during a temperature inversion. A similar feature was identifiable during another depletion episode on the early morning of 14 September 2010 (Figure 10a), with very low wind speeds at the low measurement levels and mild turbulent conditions at the high measurement levels (Figure 10c). In contrast, calm conditions were present during the full-profile ozone depletion event observed on October 2006 (Figure 10i).

Humid conditions have been previously reported to favour the occurrence of low ozone concentrations (Altimir et al., 2006). During both near-surface and full-profile ozone depletion events, the relative humidity was high (Figure 10 b & h). The results obtained here thus indicate that the ozone depletion at this relatively clean boreal forest site tends to occur under low-mixing calm conditions with a high moisture content in the air (as can be seen in Figure 10 c, d, i & j). Similar low-mixing conditions have been observed previously in association with polar boundary layer ozone depletion events (Simpson et al., 2007; Cao et al., 2016), but no relation to relative humidity, as found in our observations, was reported in the polar region (note that polar ozone depletion is characterised by halogen chemistry, which has not been observed at our boreal forest site).

In addition, it seems that an elevated CO concentration is a potential indicator of a full-profile depletion (Figure 10k), since low ozone concentrations were found in association with high CO concentrations. The recovery of ozone concentrations from the depletion event was observed first at high measurement levels, and at about the same time the CO concentration began to decrease. However, no clear trace of such a feature was seen in the near-surface depletion case, though the highest CO concentrations seemed to coincide with the lowest ozone concentrations (Figure 10e). The relation between the ozone depletion and NOx concentration was also inspected (Figure 10 f & l), but no systematic connection that could explain our observations was found.

## 4. Conclusions

Here, we investigated vertical profiles of ozone concentrations inside and above a conifer stand over a period of 20 years (1996 - 2016) in a boreal forest environment in Southern Finland (the SMEAR II station). The variations in ozone concentrations were investigated in relation to other measured trace gases and meteorological variables. Our results showed that, in general, ozone concentrations increased with an increasing height above the ground. The daily median ozone concentrations at our measurement site were found to in the range of about 20 – 50 ppb, with clear maxima in the springtime.

Temporal significant decreases of ozone concentrations at the ground level have been observed in very polluted urban environments (US Nat. Res. 1992; Gregg et al., 2003). However, to our knowledge and based on observations, the occurrence of very low ozone concentrations in relatively clean environments other than the polar regions has not been reported previously. The main aim of our study was to find out the frequency and strength of ozone depletion events at our boreal forest site. By analysing 20 years of ozone measurements, we identified on average more than a thousand of 10-min-periods at 4.2 m, during which ozone concentrations were below 10 ppb, and a few tens of cases of ozone concentration below 2 ppb. These low ozone concentrations episodes occurred typically at low temperatures, low wind speeds, high relative humidities and limited intensity of solar radiation, which are conditions likely related to the formation of a shallow mixing layer. When such conditions prevail, neither photochemical production nor enrichment from the residual layer can provide sufficient replenishment of ozone into the confined mixing layer, while ozone loss through deposition and chemical consumption continued unperturbed.

Future investigations should relate ozone depletion to ozone chemistry and boundary layer dynamics in order to understand the underlying mechanisms involved in these particular events. In addition, it is worth exploring whether there is a connection between ozone depletion and atmospheric new particle formation. Anyhow, several unrevealed or at least rarely-observed processes are likely to take place during ozone depletion events in clean to moderately-polluted





environments.

## 5. Acknowledgements

This work was supported by the Academy of Finland Centre of Excellence (project no. 272041 and 1118615), the European
Union's Horizon 2020 research and innovation program under grant agreement no. 654109 (ACTRIS-2) and the European
Union Seventh Framework Program (FP7/2007-2013 ACTRIS under grant agreement no. 262254). Dr. Matti P. Rissanen
acknowledges the Academy of Finland (project no. 299574). Xuemeng Chen appreciates gratefully the Doctoral Programme
in Atmospheric Sciences (ATM-DP, University of Helsinki) for financial support. Lauriane L. J. Quéléver expresses
gratitude to the COALA project (grant no. 638703) funded by the European Research Council. Also help from Dr. Pasi
Kolari, Stephany Buenrostro Mazon and students participating in the 2016 Autumn School in Physics, Chemistry and
Biology of Air Pollution and their Effects are acknowledged.

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




**Tables and Figures**

Table 1. The number of ozone depletion events seen at different heights that last for more than 3 hours. Ozone depletion events were classified as concentration drops below 10, 5, 2 and 1 ppb. The statistics were based on the original 10-min data.
5  Statistics based on gap-filled data are presented in brackets. The gap-filling procedure resulted in no significant changes in the statistics of depletion events for heights of 4.2 m-67.2 m.

| Height [m] | 125 | 101 | 67.2 | 50.4 | 33.6 | 16.8 | 8.4 | 4.2 |
|---|---|---|---|---|---|---|---|---|
| < 10 ppb | 0 (6) | 0 (10) | 148 (161) | 141 (185) | 178 (192) | 202 (215) | 241 (256) | 356 (378) |
| < 5 ppb | 0 (1) | 0 (1) | 26 | 28 (32) | 29 (31) | 35 (37) | 43 (47) | 63 (67) |
| < 2 ppb | 0 | 0 | 4 | 5 | 6 | 5 | 6 | 6 |
| < 1 ppb | 0 | 0 | 1 | 1 | 1 | 1 | 2 | 2 |

10  Table 2. The number of full-profile and near-surface ozone depletion events that last for more than 3 hours. Since the two highest measurement levels (101 m and 125 m) were introduced in February 2013 only, the statistics for the full-profile depletion events shown here are based on measurements at 67.2 m and lower heights. The statistics were based on the original 10-min data. Statistics based on gap-filled data are presented in brackets.

| Depletion event types | Full-profile[*] | Near-surface[**] | | | | |
|---|---|---|---|---|---|---|
| Height [m] | 67.2 | 50.4 | 33.6 | 16.8 | 8.4 | 4.2 |
| < 10 ppb | 120 (125) | 0 | 7 | 26 | 98 (100) | 160 (164) |
| < 5 ppb | 26 | 0 | 1 | 3 | 6 | 13 (14) |
| < 2 ppb | 4 | 0 | 0 | 0 | 1 | 1 |
| < 1 ppb | 1 | 0 | 0 | 0 | 0 | 0 |

[*]Full-profile ozone depletion events here are determined using the 3rd quartile of the difference in the ozone concentration between 67.2 m
15  and 4.2 m, which has a value smaller than 2 ppb.
[**]Near-surface ozone depletion events here are determined using the 1st quartile of the difference in the ozone concentration between 67.2 m and the height that the depletion reaches, which has a value larger than 2 ppb. Meanwhile, the 3rd quartile of the ozone concentration measured at 67.2 m during the depletion event is larger than 10 ppb.





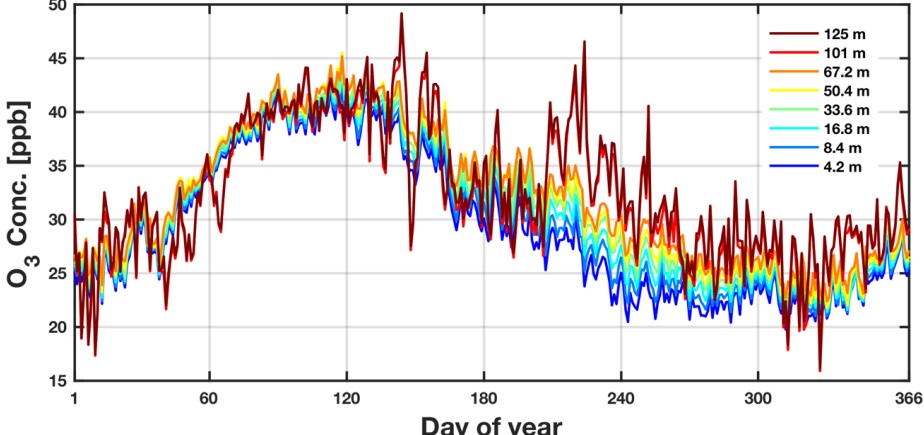

Figure 1. Daily medians of ozone concentrations at different heights for years 1996-2016. Heights are indicated by different colours.





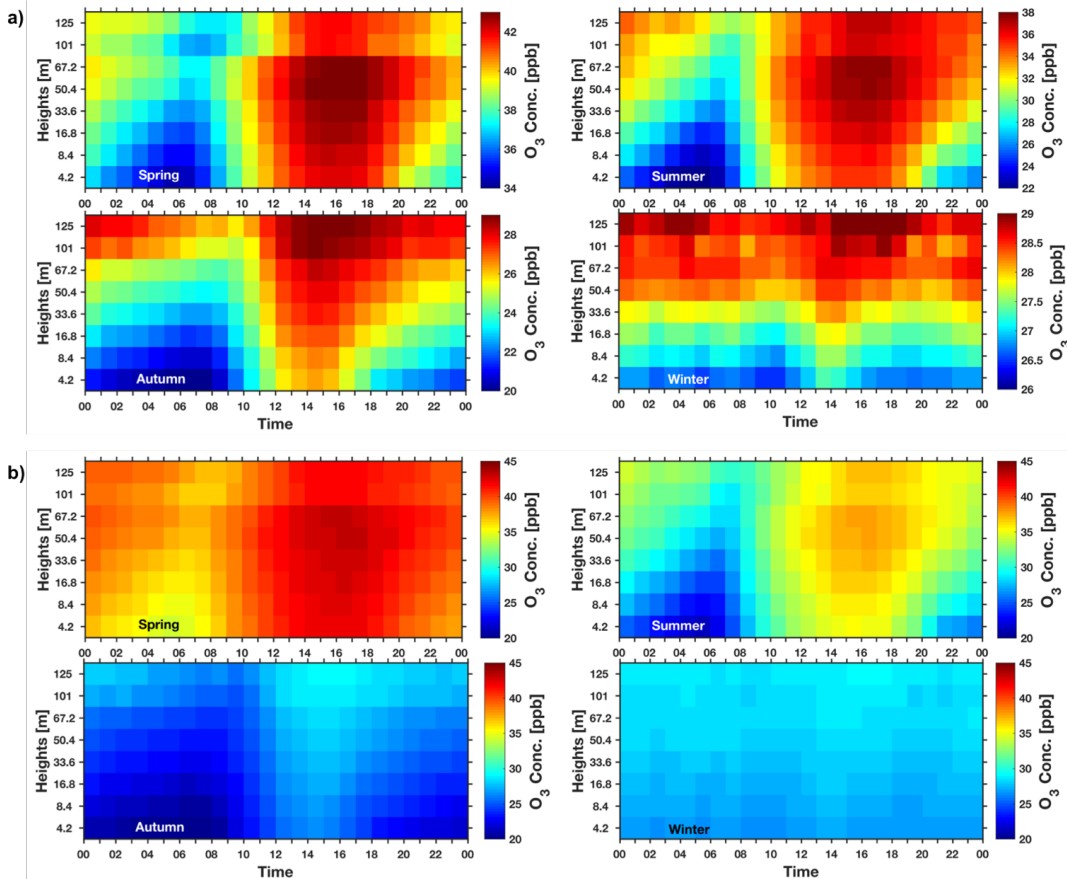

Figure 2. Diurnal profiles of ozone concentrations in different seasons. Spring: March-May, summer: June-August, autumn: September-November, and winter: December-February. Panel a) ozone concentrations in different scales for different seasons to emphasise the profile information. Note the significantly larger variability of O3 concentrations during summer months. Penal b) ozone concentrations in the same scale for different seasons.





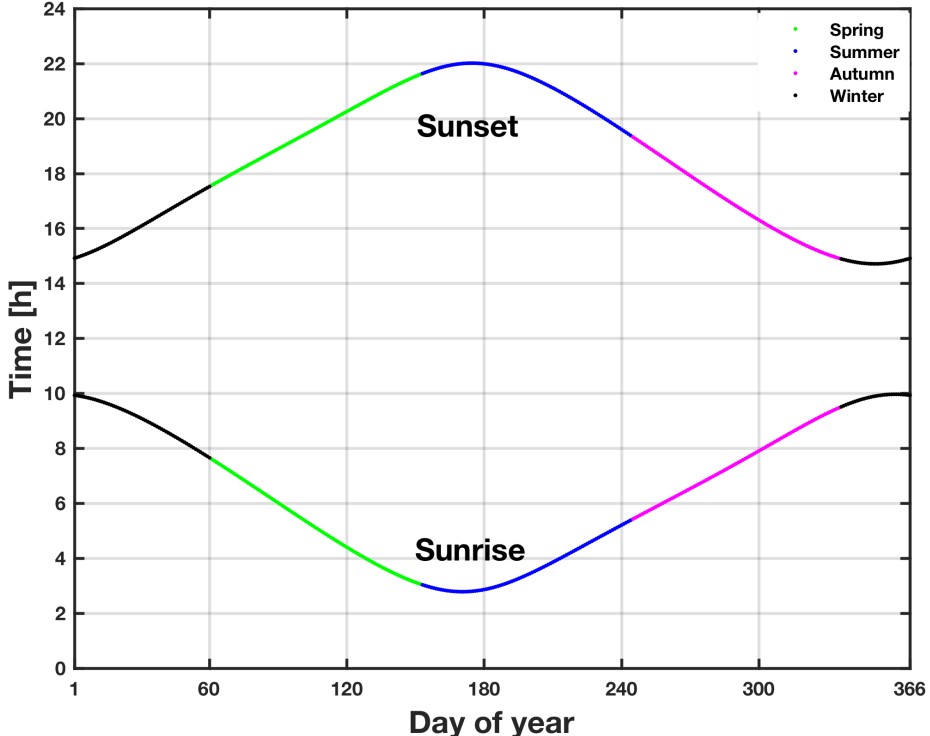

Figure 3. Estimated sunrise and sunset time of different seasons for Hyytiälä SMEAR II station (61°51'N, 24°17'E, 181 meters above sea level).





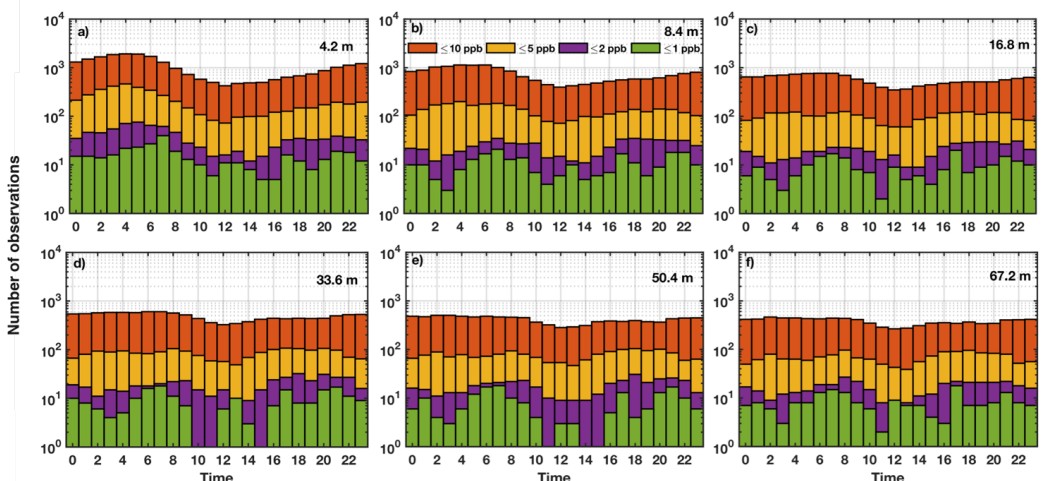

Figure 4. Hourly distribution of observations of ozone concentration drops below 10, 5, 2 and 1 ppb at different heights: a) 4.2 m, b) 8.4 m, c) 16.8 m, d) 33.6 m, e) 50.4 m, and f) 67.2 m. The statistics were based on the 10-min ozone data and the observation frequency is presented in the logarithmic scale.




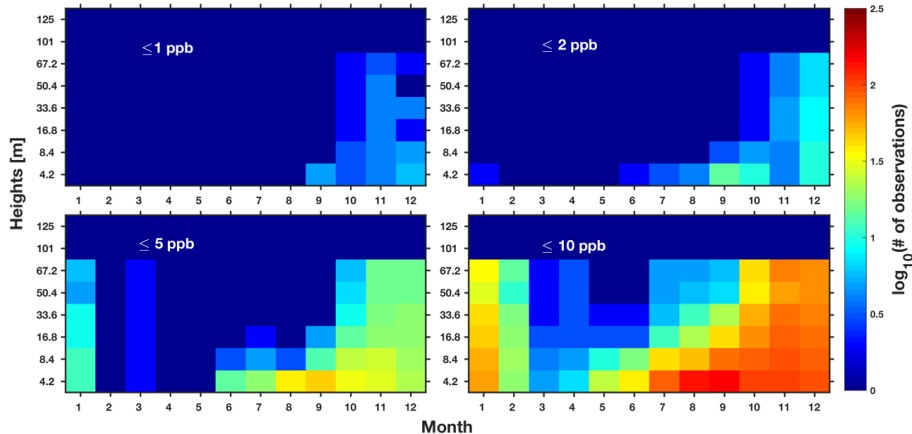

Figure 5: Vertical extension of ozone depletion events that last longer than 30 min. The numbers of depletion event observations are colour coded in logarithmic scale for ozone concentration drops below 10, 5, 2 and 1 ppb.





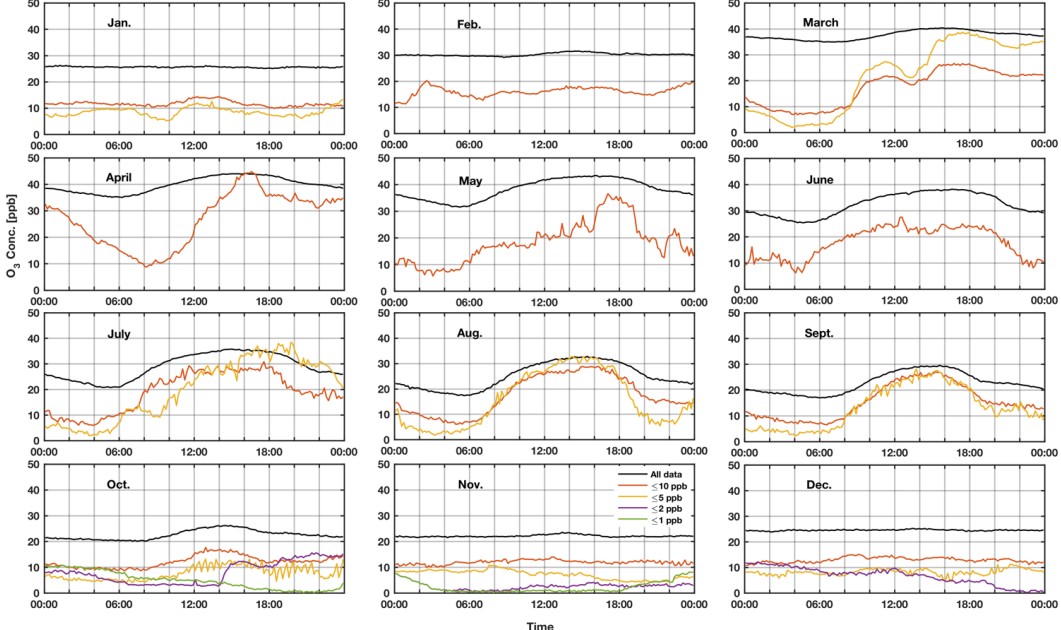

Figure 6: Diurnal patterns in median ozone concentrations at 4.2 m in different months, from all the data (black), from below-10-ppb depletion events (red), from below-5-ppb depletion events (orange), from below-2-ppb (purple) and from below-1-ppb (green). A minimum duration threshold of 3 hours was used to select days with ozone depletion events, based on the gap-filled data.





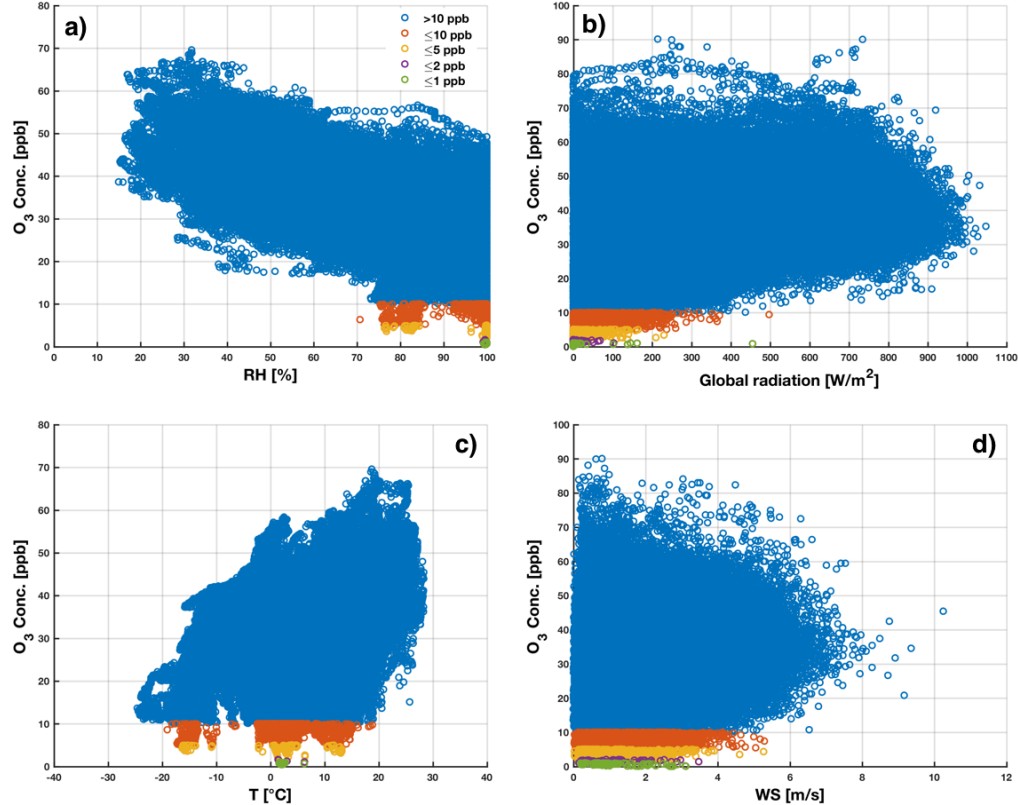

Figure 7. Ozone concentrations vs. a) relative humidity (RH), b) global radiation, c) temperature (T), and d) wind speed. Data were from the 16.8-m measurement level, with an exception of the global radiation data from 18 m.





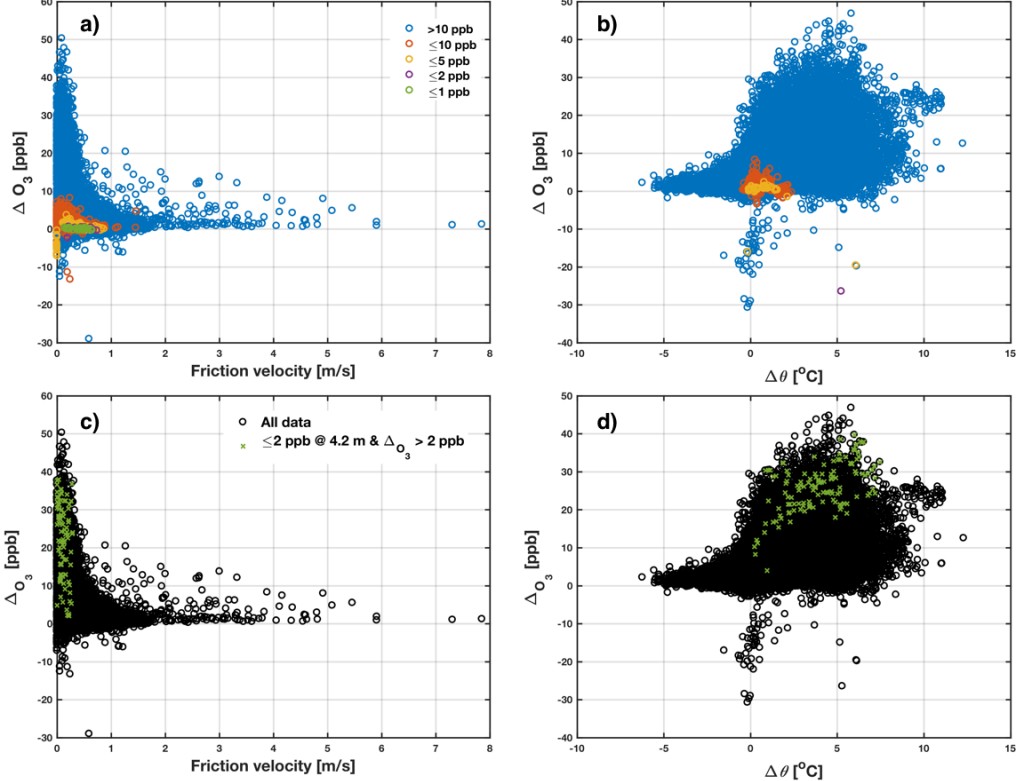

Figure 8. Ozone concentration difference between 67.2-m and 4.2-m measurement levels vs. a)&c) friction velocity determined at 23.3 m and b)&d) potential temperature difference between 67.2-m and 4.2-m measurement levels. The colour codes in a) and b) represent concentration drops below 10, 5, 2 and 1 ppb from the ozone measurement at 67.2 m. The green crosses in c) and d) mark the data points fulfilling the conditions that the ozone concentration at 4.2 m is smaller than 2 ppb while the difference in the ozone concentration between 67.2 m and 4.2 m is larger than 2 ppb.





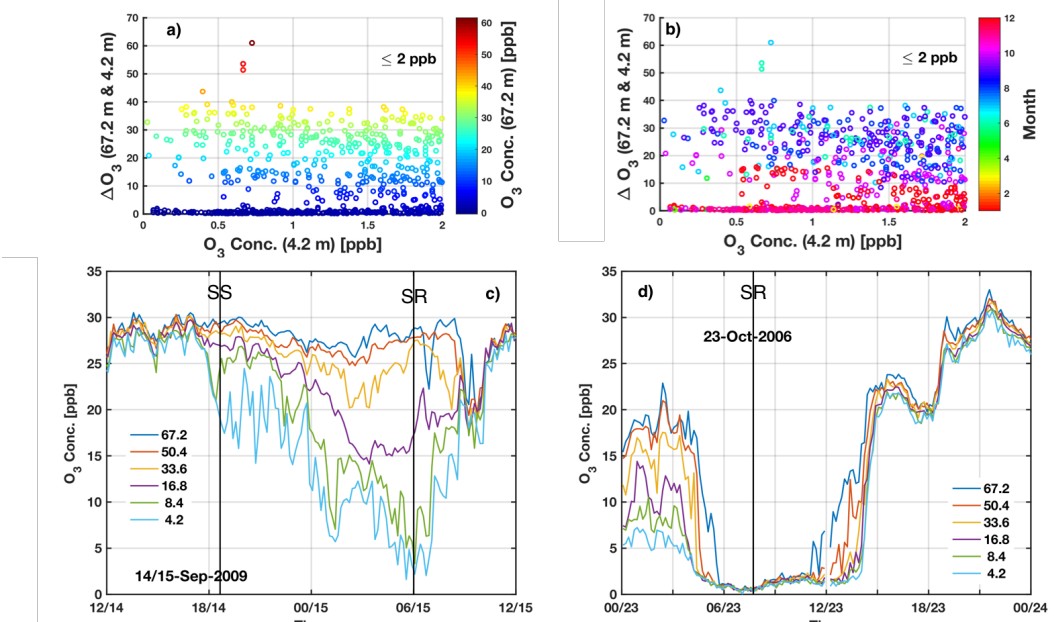

Figure 9. The ozone concentration difference between 67.2 and 4.2 m vs. the ozone concentration (≤ 2 ppb) at 4.2 m, a) colour-coded by ozone concentrations at 67.2 m and b) colour-coded by month. c) A near-surface depletion event on September 14-15[th], 2009. d) A full-profile depletion event on October 23[rd], 2006. The vertical black lines indicate the sunrise (SR) and sunset (SS) times.







Figure 10. The influences of a) & g) air temperatures ($T$), b) & h) relative humidity (RH), c) & i) wind speeds, d) & j) carbon dioxide ($CO_2$) concentrations e) & k) carbon monoxide (CO) concentrations and f) & l) nitrogen oxides (NOx) concentrations on the evolvement of the ozone concentration along time. Left panel: a near-surface depletion event on September 14-15[th], 2009. Right panel: a full-profile depletion event on October 23[rd], 2006.

