# Peer review of "Observations of ozone depletion events in a Finnish boreal forest"

_Atmospheric Chemistry and Physics, 2017_

## Referee Comment (RC1) · Anonymous Referee #1 · 23 Jul 2017

This study interrogates an impressive long-term 20-year (1996-2016) high temporal resolution dataset of vertical ozone concentration profiles within and above a canopy in a boreal forest in southern Finland. The analyses focus on understanding and characterizing "ozone depletion events", when the ozone concentration drops below 10 ppbv (and even lower) for > sustained 30-minutes period. The study further examines relationships between key meteorological and some associated chemical variables to gain further insight into the underlying mechanisms of the ozone behaviour in and around the canopy. The statistical analyses are rigorous and solid. The Figures are clearly presented and readable. The paper deserves to be published in Atmospheric Chemistry and Physics. I have some further comments and questions below that are mostly to stimulate further discussion.

1. The depletion events occur in Autumn and Winter when the temperature is low and relative humidity is high. No plant photosynthesis/production occurs during this period (i.e. no stomatal uptake of ozone). Also no substantial BVOC emissions occur during the winter at cold T. The ozone depletion events appear to be mostly driven by dynamics under the shallow winter boundary layer conditions. The ozone must be sticking to the plant and soil surfaces and draining away. Is there snow on the ground during the periods? Why does the human population care about these ozone depletion events in the winter? The ozone concentrations are actually quite high in and around the canopy in the spring and summer.

2. The median ozone concentrations at all levels are rather high in spring (∼40 ppbv) and summer (∼35 ppbv). Conventional knowledge has indicated that the ozone concentrations should be lower than these values in "pristine" forested environments. These values suggest that there is net production of ozone occurring under the forest conditions. Are the BVOC emissions contributing to this production? People generally believe, perhaps wrongly, that BVOCs in "pristine" forested environments reduce ozone levels by direct reaction. This dataset seems to suggest otherwise? Can we learn anything about the preindustrial atmosphere from these results?

3. I am surprised by the measured NOx concentrations. It is somewhat difficult to see exactly from Figure 10, but it looks like NOx is in 1-3 ppbv range increasing to 5ppbv in sporadic dynamical events. Where do these relatively high NOx levels come from in this environment?

4. Figure 7 is intriguing. Ozone concentrations have a very strong positive relationship with temperature, but not so much with global shortwave radiation. The higher ozone concentrations show relatively limited relationship with global radiation, while the lower ozone concentrations show a weak positive trend with increasing shortwave. What is the cause of this behaviour?

5. Some other previous studies do report very low measured ozone concentrations in

the Amazon forest canopy, which certain global models are unable to reproduce (e.g. Pacifico et al., ACP, 2015).

6. In the past 20 years, massive global environmental change has taken place. Is there any evidence for trends in the ozone concentrations and their drivers across the past 2 decades?

7. Page 1, Line 38: "The loss processes of tropospheric ozone include dry and wet deposition". Please correct. Ozone itself does not wet deposit. Ozone precursors do wet deposit.

---

## Referee Comment (RC2) · Anonymous Referee #2 · 24 Jul 2017

Summary

This study provides a very detailed analysis of a long term (about 20 years) dataset in order to gain a better understanding of ozone depletion events in a 'pristine' Boreal forest environment. Detailed chemical and meteorological datasets are analysed to highlight the key processes that contribute to such events on both a diurnal and vertical profile basis. Overall it is indicated that dynamics play a very strong role (E.g. low temperature, high humidity and stable conditions) in these depletion events which typically occur during autumn and winter. This therefore suggests the meteorological conditions act to enhance ozone deposition which is not able to be replaced by photochemical production and other chemical processes. This paper is very well written and clear and provides a very useful addition to the literature. Therefore I feel it should be

published in ACP after the following very minor comments are addressed.

Specific Comments:

Section 3.3 first paragraph: Figure 7 appears to show a positive correlation between ozone and temperature and a strong negative correlation between humidity and ozone. It would be quiet interesting to see a plot of temperature versus humidity to see the trends between these variables. This could serve as an insight into the potential role stomatal closure of plants plays in the ozone depletion events. E.g. Lower temperatures and higher humidities tend to lead to plants opening stomata and thus enhancing ozone deposition leading to lower concentrations. The authors go onto showing the impacts of both RH and temperature together in Figure 10 for a couple of isolated events but it would be interesting to see the trend in the combined effect of RH and temperature over all events to see if this can explain any potential changes stomatal conductance could have on deposition and hence concentrations.

Section 3.3. final paragraph, page 8 lines 15-20. The authors briefly touch on the influence of CO and NOx concentrations have on the specific ozone depletion events that they present in Figure 10. Although no clear trend is shown for the first event (Figure 10e and f), for the full depletion event (Figure 10 k and l) it is shown that both NOx and CO concentrations are high during the event and CO in particular is high leading up to the ozone depletion period. These values are a lot higher than is expected in typically clean forest environments. The location of the site is to the North/North East of two large cities (Tampere and Helsinki) and therefore perhaps regional transport of pollutants could be playing a role? I am thinking mainly in the terms of transport of high NOx which could contribute to ozone titration and enhanced depletion. Could this be a potential chemical pathway to the ozone depletion events?

Minor Comments:

Page 4, Line 33: Please correct extend to 'extent'

References: Please ensure that all references have associated DOIs.

---

## Author Comment (AC1) · 13 Oct 2017

We appreciate the careful review provided by the referee. Below are our responses to the comments of the referee. Changes in the revised manuscript are marked in red and in italics in the responses.

**1. The depletion events occur in Autumn and Winter when the temperature is low and relative humidity is high. No plant photosynthesis/production occurs during this period (i.e. no stomatal uptake of ozone). Also no substantial BVOC emissions occur during the winter at cold T. The ozone depletion events appear to be mostly driven by dynamics under the shallow winter boundary layer conditions. The ozone must be sticking to the plant and soil surfaces and draining away. Is there snow on the ground during the periods? Why does the human population care about these ozone depletion events in the winter? The ozone concentrations are actually quite high in and around the canopy in the spring and summer.**

Yes. We agree with the reviewer that ozone depletions observed at our site are likely caused by ozone deposition onto surfaces. However, we cannot exclude the role of chemistry completely based on our analysis.

Snow on the ground is common in winter at our measurement site. This information is added into the revised manuscript in sections 2 and 2.3.

*Snow on the ground during winter months is common at our measurement site…*

*No screening according to snow on the ground was taken into account in our data analysis.*

O3 is one of the major oxidants in the atmosphere and the O3 depletion has the potential to significantly alter the atmospheric chemistry of the depleted region. This influences atmospheric oxidation capacity with consequences on related trace gas removal and secondary aerosol particle formation, and thus affects local and regional air quality. Therefore, it is worth investigating into this phenomenon and understanding the frequency of its occurrence.

During summer and spring, transportation from continental areas may bring additional ozone or ozone precursors to our site. We clarified in section 2 that our site can receive transported pollutants.

*… and our site can receive transported pollutants from nearby cities (e.g. Tampere and Helsinki) and continental areas (Riuttanen et al., 2013).*

**2. The median ozone concentrations at all levels are rather high in spring (40 ppbv) and summer (~35 ppbv). Conventional knowledge has indicated that the ozone concentrations should be lower than these values in "pristine" forested environments.**

**These values suggest that there is net production of ozone occurring under the forest conditions. Are the BVOC emissions contributing to this production? People generally believe, perhaps wrongly, that BVOCs in "pristine" forested environments reduce ozone levels by direct reaction. This dataset seems to suggest otherwise? Can we learn anything about the preindustrial atmosphere from these results?**

BVOCs may contribute to ozone production at the presence of NOx. Our measurement site cannot be considered as a 'pristine' environment, because we certainly receive transported pollutants, such as NOx and O3 (Riuttanen et al., 2013). By carefully selecting data from the clean sectors based on air mass trajectory analysis, it might be possible to get some idea about preindustrial conditions, but we are afraid that such an analysis would substantially complicate our data treatment and is, therefore, out of the scope of this paper.

**3. I am surprised by the measured NOx concentrations. It is somewhat difficult to see exactly from Figure 10, but it looks like NOx is in 1-3 ppbv range increasing to 5ppbv in sporadic dynamical events. Where do these relatively high NOx levels come from in this environment?**

These relatively high NOx may be caused by transported sources emitted from nearby cities and from continental areas (e.g.(Riuttanen et al., 2013)).

**4. Figure 7 is intriguing. Ozone concentrations have a very strong positive relationship with temperature, but not so much with global shortwave radiation. The higher ozone concentrations show relatively limited relationship with global radiation, while the lower ozone concentrations show a weak positive trend with increasing shortwave. What is the cause of this behaviour?**

Figure 7 is based on all 10-min data collected during 1996-2016. The pattern between global radiation and ozone concentrations can be complicated by seasonal and diurnal variations in meteorological parameters, boundary layer heights as well as physical, chemical and biological sinks for ozone. We apologise that the figure caption was not clear enough and it is improved in the revised manuscript.

*Figure7. Ozone concentrations vs. a) relative humidity (RH), b) global radiation, c) temperature (T), and d) wind speed. Plots are made with 10-min data collected from the 16.8-m measurement level during 1996-2016, with an exception of the global radiation data from 18 m.*

**5. Some other previous studies do report very low measured ozone concentrations in the Amazon forest canopy, which certain global models are unable to reproduce (e.g. Pacifico et al., ACP, 2015).**

We thank the reviewer for this valuable comment. Indeed Pacifico et al. (2015) have observed low ozone concentrations in the Amazon forest. However, their measurements of ozone were carried out at one single height: 5 m above ground level (agl) in Porto Velho and 39 m agl in Cuieiras forest reserve. An article cited by Pacifico et al reported a gradient observed in ozone concentrations at another Amazonian cite: the ozone concentration was about 15 ppb at 50 m, which decreased down to below 5 ppb as moving to the ground level (Rummel et al., 2007). Although low ozone concentrations were reported by Pacifico et al. (2015) and Rummel et al. (2007), their ozone data showed an expected diurnal cycle, being high during daytime as a result of photochemical production, which is different from our focus in this manuscript – ozone depletion events. However, we added citations to Pacifico et al. (2015) and Rummel et al. (2005) in the introduction to point out that low surface ozone concentrations have been reported also in the Amazonian forest.

*Also low surface ozone concentrations were reported at different sites in the Amazonian forest (Rummel et al., 2007; Pacifico et al., 2015).*

**6. In the past 20 years, massive global environmental change has taken place. Is there any evidence for trends in the ozone concentrations and their drivers across the past 2 decades?**

In our ozone data, no clear trend could be observed. Please see the following figure. We made boxplots for daily median ozone concentrations over 1996-2016. Data collected at different heights were plotted into different subfigures. In 1996 and 2016, data were available only for half of the year. Data were all from the first half of the year in 2016 whereas data were mainly from the latter half of the year in 1996. Without these two years taken into account, no very clear trend is identifiable.

[Figure]

[Figure]

**7. Page 1, Line 38: "The loss processes of tropospheric ozone include dry and wet deposition". Please correct. Ozone itself does not wet deposit. Ozone precursors do wet deposit.**

We appreciate that the reviewer spotted this out. The sentence is modified as

*The tropospheric ozone is reduced directly by dry deposition, photolysis and reactions with other trace gases and indirectly via wet deposition of ozone precursors. The dry deposition of ozone can be enhanced under moist conditions (Fuentes et al., 1992).*

**References:**

Fuentes, J. D., Gillespie, T. J., Hartog, G. d., and Neumann, H. H.: Ozone deposition onto a deciduous forest during dry and wet conditions, Agricultural and Forest Meteorology, 62, 1-18, 1992.

Pacifico, F., Folberth, G. A., Sitch, S., Haywood, J. M., Rizzo, L. V., Malavelle, F. F., and Artaxo, P.: Biomass burning related ozone damage on vegetation over the Amazon forest: a model sensitivity study, Atmospheric Chemistry and Physics, 15, 2791-2804, 10.5194/acp-15-2791-2015, 2015.

Riuttanen, L., Hulkkonen, M., Dal Maso, M., Junninen, H., and Kulmala, M.: Trajectory analysis of atmospheric transport of fine particles, $SO_2$, $NO_x$ and $O_3$ to the SMEAR II station in Finland in 1996-2008, Atmos. Chem. Phys., 13, 2153-2164, 10.5194/acp-13-2153-2013, 2013.

Rummel, U., Ammann, C., Kirkman, G. A., Moura, M. A. L., Foken, T., Andreae, M. O., and Meixner, F. X.: Seasonal variation of ozone deposition to a tropical rain forest in southwest Amazonia, Atmos. Chem. Phys., 7, 5415–5435, 2007.

---

## Author Comment (AC2)

We appreciate the careful review provided by the referee. Below are our responses to the comments of the referee. Changes in the revised manuscript are marked in red and in italics in the responses.

**Specific Comments:**

**Section 3.3 first paragraph: Figure 7 appears to show a positive correlation between ozone and temperature and a strong negative correlation between humidity and ozone. It would be quiet interesting to see a plot of temperature versus humidity to see the trends between these variables. This could serve as an insight into the potential role stomatal closure of plants plays in the ozone depletion events. E.g. Lower tempera- tures and higher humidities tend to lead to plants opening stomata and thus enhancing ozone deposition leading to lower concentrations. The authors go onto showing the impacts of both RH and temperature together in Figure 10 for a couple of isolated events but it would be interesting to see the trend in the combined effect of RH and temperature over all events to see if this can explain any potential changes stomatal conductance could have on deposition and hence concentrations.**

We appreciate the comments and suggestions from the reviewer! We made a T vs. RH plot to outline their synergic effect during the low ozone concentration episodes (see the figure below). The high RH and low T may favour stomata opening for biological uptake of ozone directly via stomata. However, many results show that wet surfaces can form an additional, non-stomatal sink for ozone, which is adsorbed to the water film formed on plant surface (Fuentes et al., 1992; Altimir et al., 2006). The high humidity ozone depletion events are thus probably linked to a combination of stomatal and non-stomatal behaviours. We clarified this in the manuscript in section 3.3.

*At RH levels higher than 70% almost all surfaces of foliage are covered with thin water film, which efficiently adsorbs ozone and decouples the stomatal conductance from O3 fluxes, forming a significant non-stomatal sink for ozone (Altimir et al 2006). This can be responsible for about 40-60% of the observed decrease in O3 concentrations in the very humid cases in our data, and we can conclude that moisture content in the air is likely to be one of the key factors controlling these low ozone episodes.*

We did not add the T vs. RH plot into the revised manuscript, because we think the information contained in this plot can be deduced from the Fig. 7 and this figure cannot add additional values to the manuscript. However, if the editor thinks it is necessary, we can have this plot as a supplementary figure.

[Figure]

We also had a look at the T and RH during depletion events. By plotting T and RH during all 26 full profile events below 5 ppb together, we could observe a general negative relationship between O3 and RH (see the figure below). But for T below -15 C, the relationship seems to reverse so that the O3 concentration shows an increase with an increasing RH. The effect of T is not as clear as expected, probably due to the large seasonal variations in T. Again, the stomata uptake of O3 under high RH conditions can be expected to be stronger, but the non-stomatal update of ozone can also be enhanced under moist conditions (Altimir et al., 2006). The reason behind the ozone depletion we observed is certainly interesting to find out, which however, requires further studies to quantify the relative roles of physical, chemical and biological loss mechanism of ozone in these phenomena.

[Figure]

**Section 3.3. final paragraph, page 8 lines 15-20. The authors briefly touch on the influence of CO and NOx concentrations have on the specific ozone depletion events that they present in Figure 10. Although no clear trend is shown for the first event (Figure 10e and f), for the full depletion event (Figure 10 k and l) it is shown that both NOx and CO concentrations are high during the event and CO in particular is high leading up to the ozone depletion period. These values are a lot higher than is expected in typically clean forest environments. The location of the site is to the North/North East of two large cities (Tampere and Helsinki) and therefore perhaps regional transport of pollutants could be playing a role? I am thinking mainly in the terms of transport of high NOx which could contribute to ozone titration and enhanced depletion. Could this be a potential chemical pathway to the ozone depletion events?**

Yes, our site can be influenced by pollution transported from nearby cities. This is clarified in the revised manuscript in section 2.

*...and our site can receive transported pollutants from nearby cities (e.g. Tampere and Helsinki) and continental areas (Riuttanen et al., 2013)*

And the reaction of NOx with O3 is a known chemical pathway that could decrease local ozone concentrations. However, by a look at our case studies shown in Figure

10, in both the near-surface and full-profile depletion events, the NOx level is rather low and largely unaffected at the onset of the depletion. While in the near-surface event not much change in NOx is observed during the whole episode, in the full-profile event the NOx level is seen to increase steadily from a low to a high value in relation to the measurement site, and retain the high value after the depletion event has passed. In light of these findings it seems that NOx cannot explain the observed O3 depletion phenomena. To ultimately reveal the potential role of NOx, and also CO, in ozone depletion, dedicated laboratory experiments are required to solve these issues.

**Minor Comments:**

**Page 4, Line 33: Please correct extend to 'extent'**

Thank you for spotting this typo out! It is corrected as suggested.

**References: Please ensure that all references have associated DOIs.**

We have added DOIs to all references that have been assigned one.

**References:**

Altimir, N., Kolari, P., Tuovinen, J.-P., Vesala, T., Bäck, J., Suni, T., Kulmala, M., and Hari, P.: Foliage surface ozone deposition: a role for surface moisture?, Biogeosciences, 3, 209-228, 2006.
Fuentes, J. D., Gillespie, T. J., Hartog, G. d., and Neumann, H. H.: Ozone deposition onto a deciduous forest during dry and wet conditions, Agricultural and Forest Meteorology, 62, 1-18, 1992.